# Structural insights into actin filament recognition by commonly used cellular actin markers

Archana Kumari[1], Shubham Kesarwani[1,2], Manjunath G Javoor[1,2], Kutti R Vinothkumar[3,*] & Minhajuddin Sirajuddin[1,**] iD

## Abstract

Cellular studies of filamentous actin (F-actin) processes commonly utilize fluorescent versions of toxins, peptides, and proteins that bind actin. While the choice of these markers has been largely based on availability and ease, there is a severe dearth of structural data for an informed judgment in employing suitable F-actin markers for a particular requirement. Here, we describe the electron cryomicroscopy structures of phalloidin, lifeAct, and utrophin bound to F-actin, providing a comprehensive high-resolution structural comparison of widely used actin markers and their influence towards F-actin. Our results show that phalloidin binding does not induce specific conformational change and lifeAct specifically recognizes closed D-loop conformation, i.e., ADP-Pi or ADP states of F-actin. The structural models aided designing of minimal utrophin and a shorter lifeAct, which can be utilized as F-actin marker. Together, our study provides a structural perspective, where the binding sites of utrophin and lifeAct overlap with majority of actin-binding proteins and thus offering an invaluable resource for researchers in choosing appropriate actin markers and generating new marker variants.

**Keywords** actin cytoskeleton; cellular markers; cryoEM; lifeAct; phalloidin
**Subject Categories** Cell Adhesion, Polarity & Cytoskeleton; Structural Biology
The EMBO Journal (2020) 39: e104006

## Introduction

The cytoskeleton protein actin can flux between globular (G-actin) and filamentous (F-actin) form, and this polymerization cycle is coupled to various cellular functions. The structure of actin is subdivided into four subdomains, SD 1–4, and contains a central pocket for ATP and magnesium ion. The polymerization kinetics of actin is dictated by the intrinsic nucleotide hydrolysis kinetics, phosphate release, and ATP turnover cycle (Pollard, 1986; Kudryashov &

Reisler, 2013). The presence and absence of γ-phosphate at the nucleotide-binding pocket induces conformational changes that leads to opening and closing of DNase-binding loop (D-loop), respectively (Kudryashov & Reisler, 2013; Merino et al, 2018; Chou & Pollard, 2019). In cells, several actin regulators can create an array of F-actin structures along with the nucleotide turnover and kinetics (Pollard, 2016), for example, stress fibers, cortical actin, lamellipodia, and filopodia (Hall, 1998); all of them show differential actin dynamics and are linked to a specialized actin-mediated cellular process. To better understand these processes, researchers often use fluorescent probes that label actin and visualize them using various microscopy methods (Melak et al, 2017). These markers broadly can be categorized into fluorescent-tagged actin, toxins, peptides, and proteins with actin-binding domains (ABDs).

Fluorescent protein variants tagged at the amino-terminus of the actin gene have been used to label actin (Belin et al, 2014; Melak et al, 2017). The advantages of this approach include measurement of actin turnover in cells as well as in vivo whole organism studies, especially in the context of the control of expression with conditional and tissue-specific promoters. The major disadvantage is the bulkiness of fluorescent proteins, which has been shown to impede incorporation of tagged G-actin into growing F-actin (Doyle & Botstein, 1996). To overcome this, fluorescent toxins that bind to F-actin such as phalloidin (Wulf et al, 1979) and jasplakinolide (SiR-actin) (Lukinavičius et al, 2014) have been employed, of which phalloidin is perhaps the most widely used. Both phalloidin and jasplakinolide stabilize F-actin (Bubb et al, 1994), and structural investigations suggest that they both bind to same actin-binding site. Of these toxins, the jasplakinolide-bound F-actin structure has been shown to mimic ADP-Pi actin transition state, i.e., an open D-loop conformation (Pospich et al, 2017; Merino et al, 2018). A similar conclusion for phalloidin could not be derived because the phalloidin-bound F-actin structures determined so far, either have myosin (Mentes et al, 2018) or filamin (Iwamoto et al, 2018) bound to the filament, both of which overlap with the D-loop region.

The other commonly used reagent for F-actin labeling is lifeAct, a 17 amino acid peptide derived from yeast actin-binding protein (Riedl et al, 2008). Since its inception, the application of lifeAct to mark actin in cells elicits polarized responses among investigators;

1 Center for Cardiovascular Biology and Diseases, Institute for Stem Cell Science and Regenerative Medicine, Bengaluru, India
2 Manipal Academy of Higher Education, Manipal, India
3 National Center for Biological Sciences, TIFR, Bengaluru, India
*Corresponding author. Tel: +91 8061948060; E-mail: vkumar@ncbs.res.in
**Corresponding author. Tel: +91 8061948133; E-mail: minhaj@instem.res.in

largely because lifeAct is shown to interfere with actin dynamics (Courtemanche et al, 2016) and it fails to label certain actin structures in cells (Munsie et al, 2009; Belin et al, 2014; Lemieux et al, 2014; Spracklen et al, 2014). LifeAct is also known to bind both G-actin and F-actin, with higher affinity towards the former form of actin (Riedl et al, 2008). However, a detailed structural analysis of lifeAct and actin interaction is still lacking.

In addition to toxins and peptides, the alternate method of actin labeling includes calponin homology domains (CH) that binds to actin, also known as tandem ABDs. The tandem CH1 and CH2 ABDs of utrophin (UTRN-ABD or UTRN-261, amino acids 1–261) have been successfully employed in F-actin visualization (Burkel et al, 2007). Biochemical, structural, and cell biological studies have proposed that the tandem arrangement of CH domain is important for F-actin binding (Winder et al, 1995; Moores & Kendrick-Jones, 2000; Galkin et al, 2002, 2003). The crystal structure and biochemical experiments carried out with peptide fragments of utrophin suggest that CH1 domain has two actin-binding sites and the third actin-binding site was proposed to be in CH2 domain (Levine et al, 1992; Keep et al, 1999). Although the CH domains have high similarity among them, the linker between CH domains is variable and it is unclear whether the tandem CH architecture is important for utrophin:actin interaction. This is supported by truncation studies, which show the CH1 domain has higher affinity similar to the UTRN-ABD and the CH2 domain is important for solubility (Singh et al, 2014). Earlier electron microscopy studies with helical reconstruction of UTRN bound to F-actin has attempted to address the importance of tandem CH domains but the details could not be delineated due to its low resolution of the maps (Moores & Kendrick-Jones, 2000; Galkin et al, 2002). Additionally, a shorter version of utrophin, UTRN-230 (amino acids 1–230) has been shown to specifically label Golgi actin (Belin et al, 2014) and nuclei actin (Du et al, 2015), further questioning the actin-binding sites and requirement of the tandem CH domain for actin interaction.

It has been well documented and acknowledged in the field that no one fluorescent actin marker is superior and all of them have certain limitations (Munsie et al, 2009; Belin et al, 2014; Courtemanche et al, 2016). Therefore, the accepted notion is that the choice of actin markers in investigations needs to be thoroughly thought through (Lemieux et al, 2014; Melak et al, 2017). However, all the studies have been limited to cell biology investigation and there is no structural study that has compared them systematically. In order to address these structural gaps, we employed electron cryomicroscopy (cryoEM) and helical reconstruction methods to determine the structures of actin markers bound to F-actin. Here, we describe the phalloidin-, lifeAct-, and utrophin-bound F-actin structures, representing toxin, peptide, and protein markers widely used in actin labeling and differences and similarities of their binding interface.

## Results

### Phalloidin-bound actin mimics the actin–ADP state

Phalloidin- and jasplakinolide-bound F-actin structures show that both share the same binding site (Bubb et al, 1994; Mentes et al, 2018; Merino et al, 2018). Jasplakinolide binding to actin induces ADP-Pi like actin conformation state with an open D-loop, a nucleotide sensing region of actin (Merino et al, 2018). The available phalloidin structures are in complex with actomyosin or actin/filamin, where both myosin and filamin binding overlaps with the D-loop (Iwamoto et al, 2018; Mentes et al, 2018). Therefore, we determined 3.8 and 3.6 Å structures of F-actin–ADP (called apo, as this has only ADP bound) and phalloidin bound using cryoEM and helical reconstruction, respectively (Fig EV1 & Table 1; Materials and Methods). Both of these F-actin structures contain ADP in the nucleotide-binding site of actin and thus form the basis for the comparison of phalloidin-induced conformational changes (Appendix Fig S1).

The cyclic heptapeptide, phalloidin adopts a wedge like conformation, and the binding pocket is buried in between three actin monomers (Fig 1A & Appendix Fig S1). Residues from the $n + 2^{nd}$ actin monomer were earlier reported to involve in hydrophobic contacts with phalloidin (Mentes et al, 2018). However, a closer inspection of the binding site reveals that the nearest residue I287 and R290 from the third actin monomer ($n + 2^{nd}$ monomer) is approximately 5 Å away from phalloidin (Fig 1B and C Appendix Fig S1E). This indicates that phalloidin mainly interacts with two actin monomers and stabilizes the filament interface (Fig 1A and B). The binding pocket contains a mixture of hydrophobic and charged residues contributing to the phalloidin binding (Fig 1B and C). Phalloidin mainly interacts with E72, H73, I75, T77, L110, N111, P112, R177, D179 of $n + 1^{st}$ actin monomer and T194, G197, Y198, S199, F200, E205, and L242 of $n^{th}$ actin monomer (Fig 1B and C Appendix Fig S1E). Superimposition of residues within the vicinity of phalloidin also does not show any major side-chain deviations between apo and phalloidin-bound structures (Fig 1C).

When the apo-, phalloidin-, and jasplakinolide-bound F-actin–ADP structures were compared, no significant structural deviation was observed except in the D-loop region (Fig 1D). In the ADP (apo) and ADP/phalloidin actin structures, the D-loop region remains in the closed state (rmsd 1.1 Å). While in the ADP/jasplakinolide-bound F-actin structure, D-loop adopts an open conformation (Merino et al, 2018) (Fig 1D), the rmsd of D-loop between jasplakinolide versus phalloidin is 2.4 Å. Since, we have determined the undecorated F-actin structure, we conclude that phalloidin binding does not induce any large conformational changes in actin and resembles the respective nucleotide state of F-actin (Fig 1D).

### LifeAct and F-actin interaction is mediated by hydrophobic contacts

From the time of its discovery, lifeAct has been widely used to detect actin using microscopy in cell biology studies (Melak et al, 2017). LifeAct is also known to influence actin dynamics and can bind both monomeric (G-actin) and F-actin (Riedl et al, 2008) but it is unclear how this occurs at molecular level. To gain structural insights into the lifeAct:actin complex, we determined a 4.2 Å structure of lifeAct-bound F-actin (Materials and Methods; Fig EV2 & Table 1). Similar to other actin markers, binding of lifeAct to F-actin does not alter the helical symmetry of the filament (Table 1).

LifeAct adopts a helical structure and binds stoichiometrically at the SD1 region of actin monomers and the carboxy-terminus of lifeAct extends toward the D-loop of the $n$-$2^{nd}$ neighboring (barbed

**Table 1. Data collection, refinement and validation statistics.**

| Sample/Parameters | F-actin–Phalloidin | F-actin–Apo | F-actin–Utrophin | F-actin–LifeAct |
|---|---|---|---|---|
| Microscope | Titan Krios G3 -X-FEG, 300 | | | |
| Voltage | 300 | | | |
| Defocus range (μ) | −1.5 to −3.0 | −1.5 to −3.0 | −1.8 to −3.3 | −1.8 to −3.5 |
| Camera | Falcon III | Falcon III | Falcon III | Falcon III |
| Pixel size (Å) | 1.38 | 1.08 | 1.38 | 1.38 |
| Total electron dose (e/Å$^2$) | 55.67 | 49.20 | 42.67 | 49.20 |
| Exposure time | 1.99 | 1.99 | 1.99 | 1.99 |
| Frames per movie | 20 | 30 | 20 | 30 |
| Number of images | 1,124 | 529 | 765 | 929 |
| 3-D refinement statistics and helical symmetry | | | | |
| Total number of helical segments extracted | 349,839 | 111,074 | 259,938 | 297,584 |
| Number of segments in map | 91,245 | 64,194 | 149,660 | 74,000 |
| Resolution (Å) | 3.6 | 3.8 | 3.6 | 4.2 |
| Helical twist | −167.02 | −166.8 | −167.34 | −166.9 |
| Rise | 27.89 | 27.25 | 28.02 | 27.44 |
| Map sharpening factor (Å$^2$) | −167.2 | −179.7 | −177.3 | −263 |
| Model composition and validation | | | | |
| Non-hydrogen atoms | 14,722 | 14,421 | 17,394 | 14,733 |
| Protein residues | 1,843 | 1,826 | 2,204 | 1,889 |
| Ligands | 5Mg, 5ADP, 3 Phalloidin | 5Mg, 5ADP | 5Mg, 5ADP | 5Mg, 5ADP |
| RMSD | | | | |
| Bond lengths(Å) | 0.012 | 0.008 | 0.006 | 0.009 |
| Bond angles (°) | 0.969 | 0.929 | 0.767 | 0.916 |
| B-factor (Å$^2$) | | | | |
| Protein | 58.50 | 51.76 | 68.95 | 79.6 |
| Actin | 58.2 | 51.5 | 56.1 | 78.2 |
| Toxin/protein[a] | 51.6 | – | 132.6 | 110.3 |
| Ligand (ADP) | 52.12 | 47.01 | 60.48 | 84.64 |
| MolProbity Score | 2.99 | 2.83 | 2.29 | 3.33 |
| Clashscore | 10.87 | 7.76 | 5.4 | 20.84 |
| Ramachandran plot:-Favored | 89.98 | 91.00 | 95.38 | 88.10 |
| Allowed | 9.79 | 8.89 | 4.52 | 11.90 |
| Outlier | 0.2 | 0.1 | 0.0 | 0.0 |
| PDB ID | 7BTI | 7BT7 | 6M5G | 7BTE |
| EMDB Code | 30179 | 30171 | 30085 | 300177 |

[a]Denotes Phalloidin, Utrophin, and LifeAct in the respective column.

end) actin monomer (Fig 2A and B). The helical nature of lifeAct allows one to orient its hydrophobic residues, V3, L6, I7, F10, and I13 toward the actin (Fig 2B). Complementing this hydrophobicity are cluster of hydrophobic residues Y143, I345, L346, L349 and M355 mediates lifeAct binding (Figs 2B and EV2D). An interesting feature is that the lifeAct-binding pocket involves D-loop residues V45, M44, and M47 of the $n$-2$^{nd}$ actin neighbor (Figs 2B and EV2D). Together these residues form a hydrophobic pocket that can accommodate the phenyl sidechain group of the F10 lifeAct peptide (Figs 2B and EV2D).

To understand the importance of the hydrophobic interface, we performed mutagenesis experiment with lifeAct-GFP. The wild-type and mutant lifeAct were expressed in U2OS cells, and their localization was imaged with actin structures (Materials and Methods). We chose V3, L6, F10, and I13 in lifeAct and replaced them with aspartic acid (Fig 2B and C). Co-localization with the SiR-actin probe showed that only residues that mediate hydrophobic contacts with F-actin as described above drastically reduced binding to F-actin in cells (Fig 2C and Appendix Fig S2), thus validating our structural observations of lifeAct and F-actin interaction. In addition, we

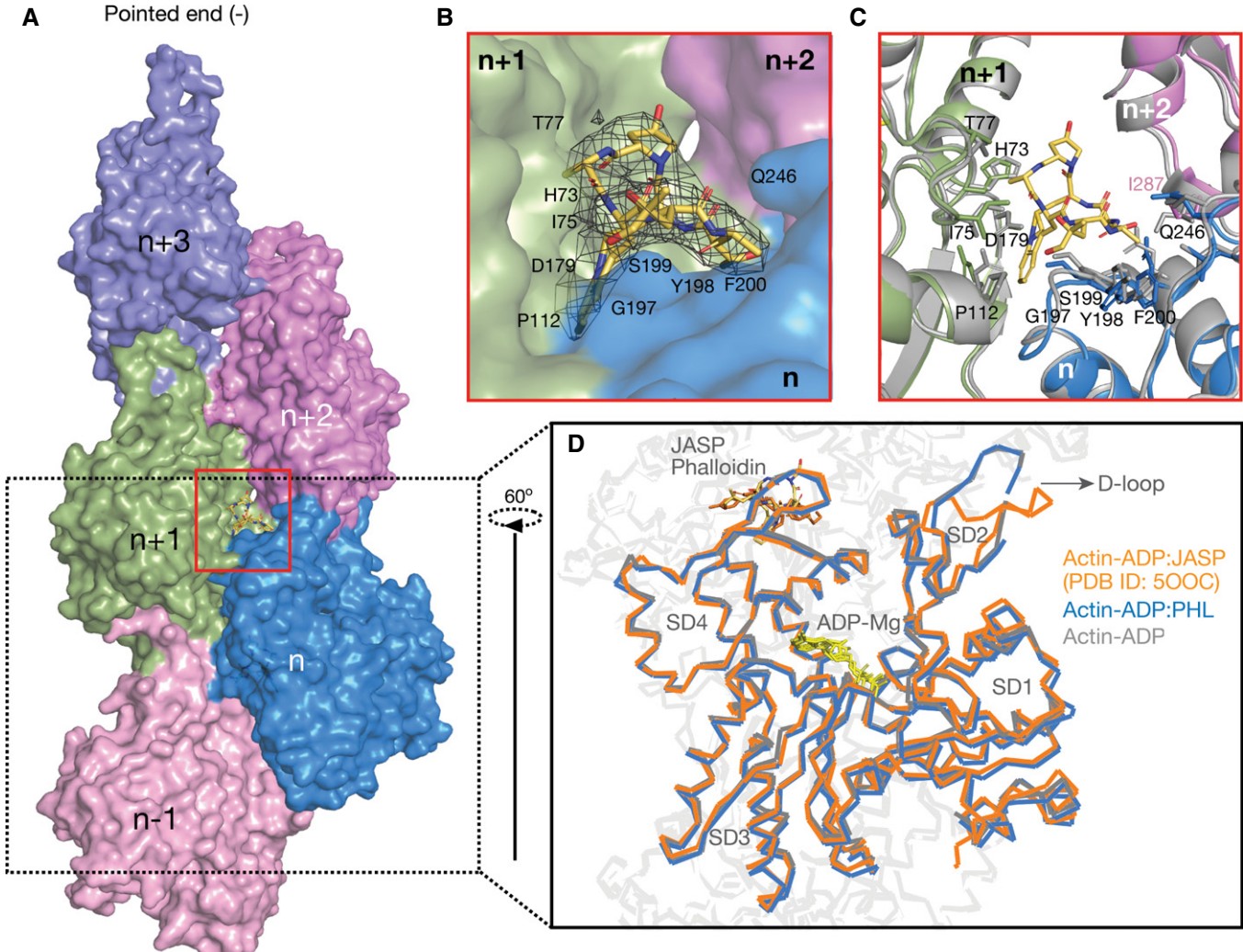

**Figure 1. Phalloidin-bound F-actin structure resembles ADP-actin state.**

A  Surface representation of F-actin–ADP model, five monomers marked as *n* series from barbed to pointed end. A phalloidin molecule (yellow stick representation) bound between three actin monomers is highlighted.

B  Expanded view of phalloidin-binding pocket as marked with red box in panel (A). The density of phalloidin from EM map is shown around the ligand.

C  Comparison of phalloidin-binding pocket residues between apo (in gray) and phalloidin bound (actin monomer colors as indicated in panel A) Key residues with their side chains and phalloidin are represented in stick representation.

D  Overlay of F-actin–ADP (gray), ADP/Phalloidin (blue), and ADP/Jasplakinolide (orange) shows the D-loop conformations across different structures as indicated.

generated a carboxy-terminal truncated version of lifeAct, called lifeAct-14 (MGVADLIKKFESIS). The lifeAct-14 labels F-actin to similar to the full-length lifeAct (lifeAct-17). Co-sedimentation assay of lifeAct-14 with F-actin also shows binding constants in similar range as the lifeAct-17 (Fig 2D). Thus, our lifeAct:F-actin complex structure provides a platform to generate newer and better lifeAct variants with desired properties.

**LifeAct senses the closed D-loop conformation**

Since lifeAct peptide binding overlaps with the D-loop of the $n$-$2^{nd}$ actin neighbor, we next probed the importance of D-loop conformation toward lifeAct and F-actin interaction. Comparison of open (jasplakinolide-bound F-actin:ADP PDB: 5OOC) versus

closed D-loop states (F-actin:ADP:lifeAct; Fig 3A) suggests that the open D-loop state is incompatible for lifeAct binding (Fig 3B). Therefore, from the structural model of lifeAct:actin complex we reasoned that lifeAct may have preference towards different biochemical states of actin. We therefore prepared two batches of F-actin, one with phalloidin bound and the other with jasplakinolide bound, representing closed and open D-loop actin state, respectively (Fig 1D). The two distinct fluorescent F-actin populations were incubated together with varying concentrations of FAM-lifeAct peptide and visualized in the same reaction chamber using TIRF microscopy (Materials and Methods; Fig 3C). At micromolar concentrations, we began to observe F-actin labeling by FAM-lifeAct; however, the co-localization was favored towards the phalloidin F-actin form (Fig 3C and D). We quantified the

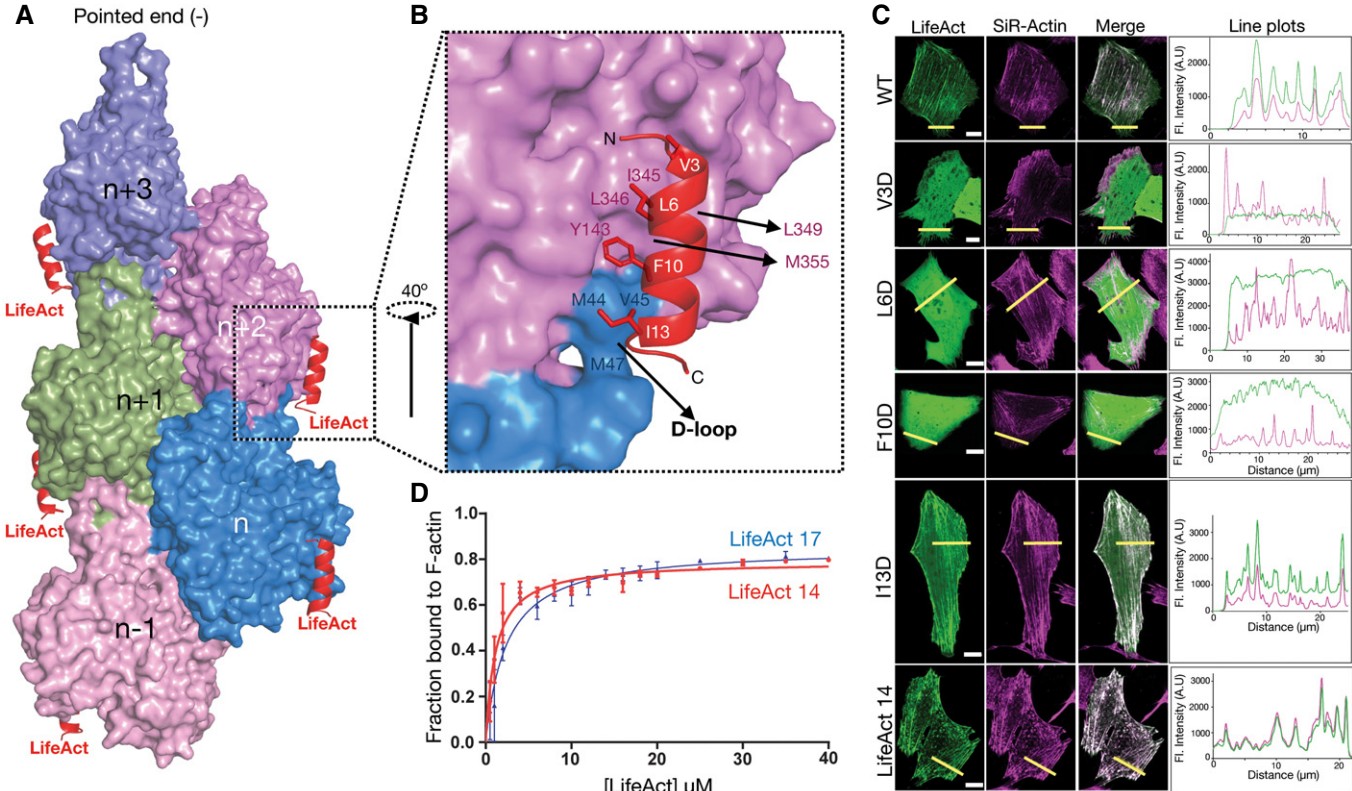

**Figure 2. Interaction of lifeAct with F-actin and mutational analysis.**

A  Surface representation of F-actin bound to lifeAct as indicated.

B  Cartoon representation of lifeAct (red) in expanded view with *n* and *n* + 2nd actin monomers (surface representation in magenta and blue), key interacting residues are highlighted.

C  Confocal images of U2OS cells transiently expressing lifeAct-GFP wild-type and mutants of lifeAct residues interacting with F-actin, cells were additionally stained with SiR-actin to confirm the actin filaments. The line scan as indicated with yellow line on the cells shows the extent of lifeAct (green) and SiR-actin (magenta) co-staining of actin structures. Scale bar = 5 μm.

D  Binding affinities calculated from titration data of co-sedimentation assays of lifeAct-14 (1.2 μM) and lifeAct-17 (2.2 μM) as indicated. Data points for each concentration were averaged from three independent experiments; error bars represent SD between independent experiments.

Source data are available online for this figure.

---

fluorescence intensity ratio of FAM-lifeAct for phalloidin versus jasplakinolide F-actin (Materials and Methods; Fig 3D) and found that a three to fourfold fluorescence increase toward the phalloidin-bound F-actin, i.e., closed D-loop conformation. This trend and the fluorescent intensity ratio were observed at different concentrations of lifeAct (Figs 3C and D, and EV3). The striking preference of phalloidin over jasplakinolide F-actin thus strongly suggests that lifeAct preferentially binds to the closed state of D-loop conformation of actin monomers in F-actin.

**The utrophin CH1 domain is sufficient for F-actin interaction**

In our quest toward structural characterization of actin markers, we then focused on the utrophin actin-binding domain, widely known as UTRN-ABD or UTRN261, amino acids 1–261. The UTRN-ABD contains two calponin homology domains (CH1 and CH2 domains), and previous structural and biochemical studies have proposed both the domains are necessary for actin interaction (Winder *et al*, 1995). The purified UTRN-ABD was used to make a complex with

F-actin (Materials and Methods), and the map was resolved to 3.6 Å resolution (Table 1 and Fig EV4). The UTRN-ABD model was built from the available X-ray structure coordinates (PDB ID: 1QAG) (Keep *et al*, 1999) and an additional amino-terminal helix (amino acids 18–33), which was partially disordered in the X-ray structure was built *de novo*. Although our cryoEM preparations contain complete UTRN-ABD protein (amino acids 1–261), in the final reconstructed map we could model only less than 50% of the utrophin, amino acids 18–135, corresponding to the CH1 domain (Figs 4A and EV4).

Previous work subdivided the CH1 domain of UTRN-ABD into ABD1 (amino acids 31–44) and ABD2 (amino acids 105–132) (Keep *et al*, 1999). From our F-actin-bound structure, we could redefine the boundaries of ABD sites; ABD1 (amino acids 18–33), ABD2 (amino acids 107–126) and a newly identified ABD site in between ABD1 and ABD2, named ABD2′ (amino acids 84–94; Fig 4B). ABD1 is the amino-terminal helix, which mainly interacts with the SD1 of the *n*th actin monomer (Fig 4A and B). Both ABD2 and ABD2′ interact with SD2, chiefly with the D-loop region of the *n*th actin

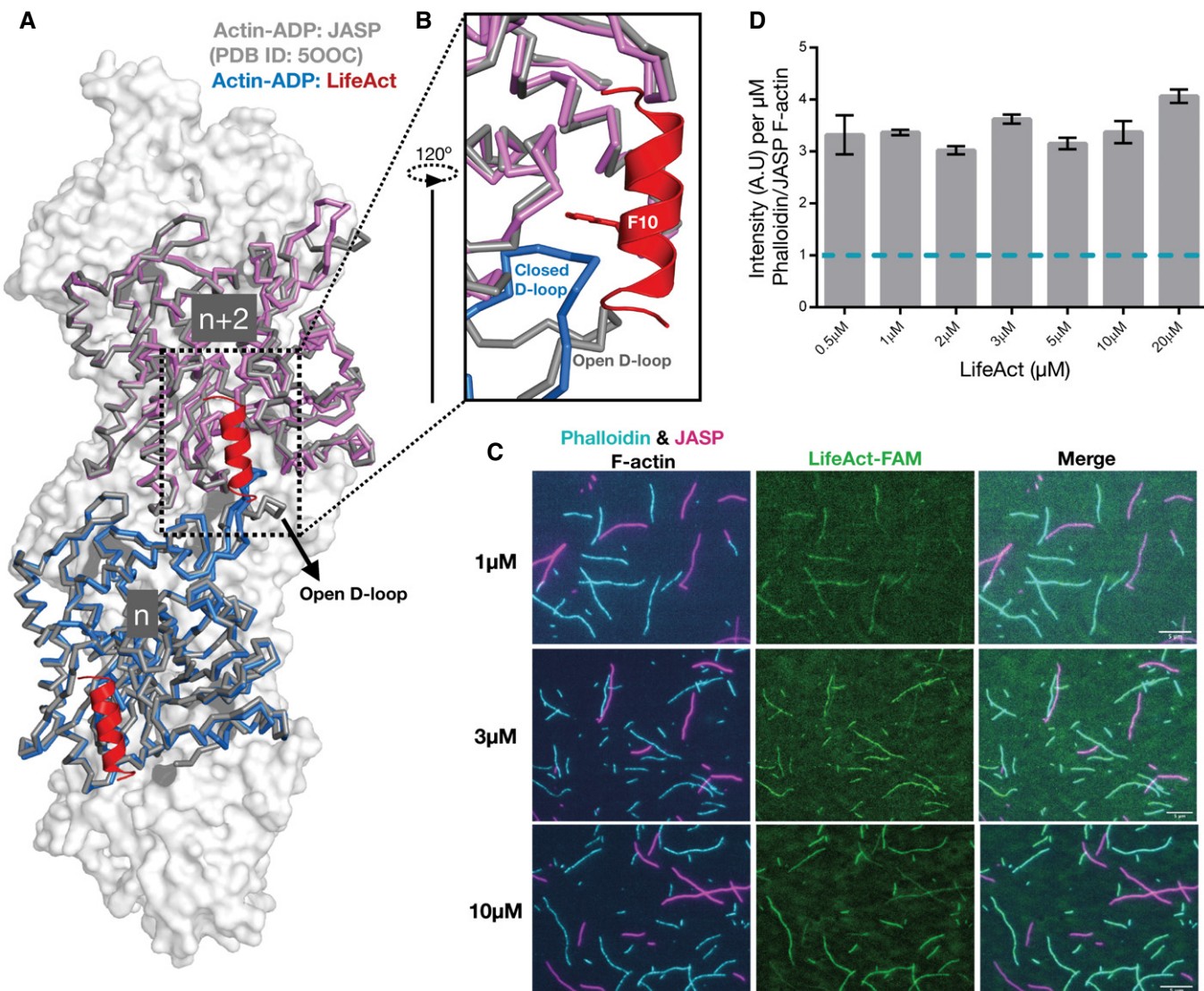

**Figure 3. LifeAct recognizes the closed D-loop state of F-actin.**

A, B  Overlay of F-actin:ADP-bound LifeAct (actin in blue and magenta ribbon representation and lifeAct in red cartoon) and jasplakinolide (in gray).

C  Representative TIRF images of lifeAct-binding experiments as indicated. The remaining lifeAct concentration images are shown in Fig EV3. Scale bar = 5 μm.

D  Mean ratio of lifeAct fluorescent intensity bound to phalloidin and jasplakinolide F-actin (mean and SEM; n = 2 or three independent experiments with > 50 actin filament for each set).

Source data are available online for this figure.

monomer, which remains in a closed conformation (Fig 4A and B). Additionally, ABD2′ is the only site that interacts with the SD1 of adjacent $n + 2^{nd}$ actin monomer (Fig 4A and B). The binding site and architecture of UTRN-ABD-CH1 are similar to the recently reported FLNaABD (Iwamoto *et al*, 2018); however, in our UTRN-ABD an additional amino-terminal helix is visible, extending toward the barbed end of the actin monomer (Fig 4A and B).

To validate our structural model of the UTRN-ABD:actin complex and the newly defined ABD1 (amino-terminal helix) and ABD2′ region, we performed mutagenesis of key interacting residues (Fig 4C–E) and co-sedimentation assays with F-actin (Materials and

Methods; Appendix Fig S3). From the ABD1 helix, we chose residues that have their side chains facing toward actin; thus, I22 makes hydrophobic contacts with P27, V30, and Y337 of actin residues, and H29 engages in a cation-pi interaction with R28 and R95 of actin (Figs 4E and EV4D). At the core of the CH1 domain (ABD2), we included N109, which makes electrostatic interactions with actin K50 and H88 side chains and the main chain carbonyl group of S52 and V54 (Fig 4C). Additionally, we included V87D from the newly identified ABD2′, which makes hydrophobic contacts with Y143 of the SD1 of the adjacent $n + 2^{nd}$ actin monomer and M44 from the D-loop of the $n^{th}$ monomer (Fig 4D). In summary, we tested the

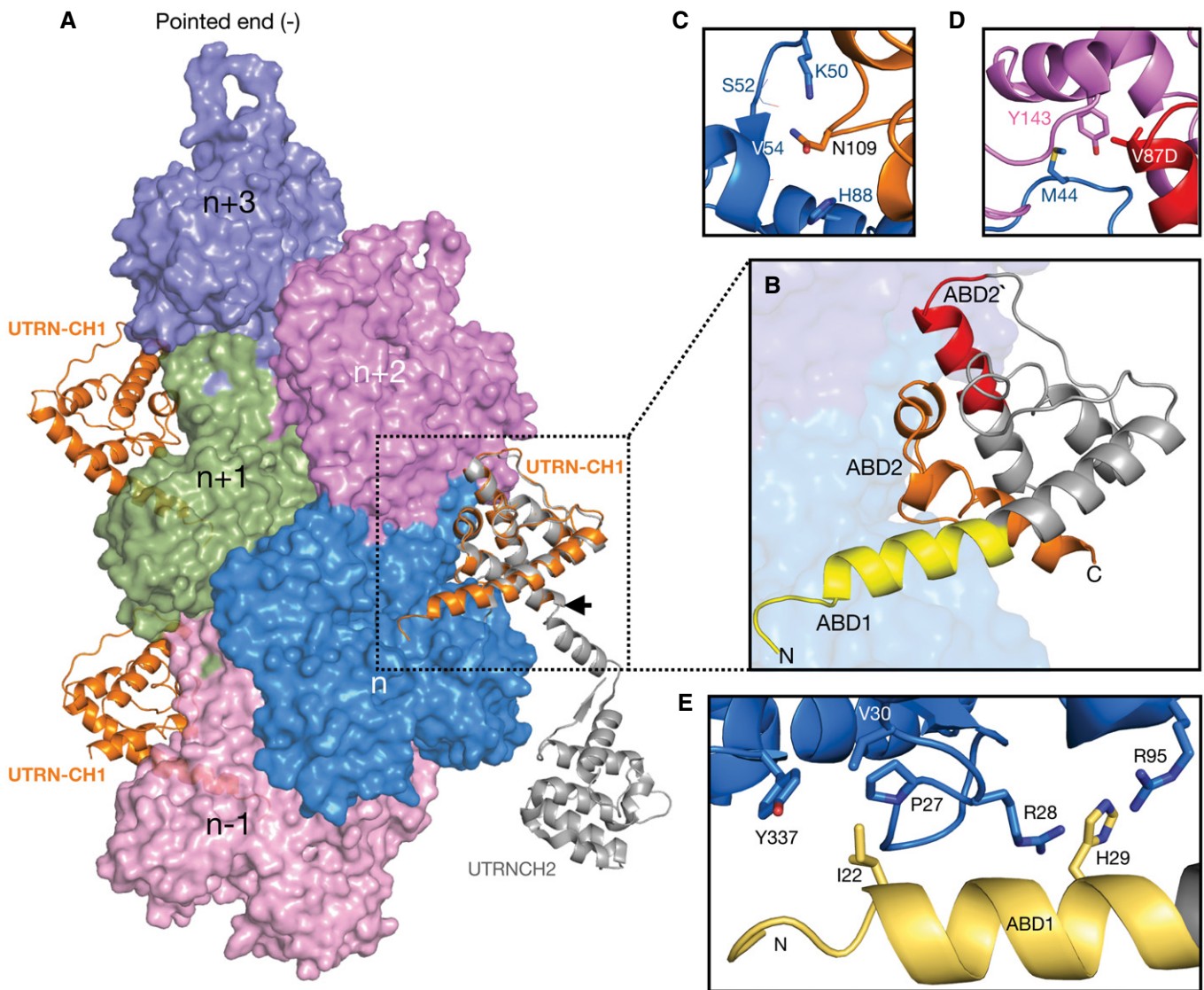

**Figure 4. Utrophin CH1 domain structure and F-actin interaction sites.**

A    Surface representation of F-actin–ADP, five monomers marked as *n* series from barbed to pointed end. The utrophin CH1 domain in orange interacts with two adjacent actin monomers thus following the actin helical pattern. The crystal structure of dystrophin/utrophin in gray (1DXX) superimposed with cryoEM utrophin CH1 model, boundary of CH1 is marked by an arrow.

B    Closer view of utrophin CH1 model, the yellow, orange, and red region depicts ABD1, ABD2, and ABD2′ sites, respectively. The ABD1 and ABD2 sites are restricted to $n^{th}$ actin monomer, and the ADB2` site partially interacts with the neighboring $n + 2^{nd}$ actin monomer.

C–E    Residual level information of key amino acids interacting with actin monomers from ABD2: N109 (C); ABD2′ site: V87 (D); and ABD1: I22, H29 (E).

following mutants, I22D and H29A from the ABD1 site, V87D and N109A for ABD2′ and ABD2 sites, respectively (Fig 4B–E).

Co-sedimentation assays of mutants compared to the wild-type UTRN-ABD protein show more than 50 and 100 times decrease in binding constants for V87D and N109A mutants, respectively (Fig 5A and B and Appendix Fig S3). The decrease in affinity by V87D and N109A mutants indicate that the core binding is mediated by the ABD2′ and ABD2 sites. Moreover, the reduced binding constants of the V87D mutant data indicates that the UTRN-ABD interacts with two neighboring ($n^{th}$ and $n + 2^{nd}$) actin monomers and thus has the ability to bind to F-actin, but not actin monomers

(Winder *et al*, 1995). Our mutation analysis also shows that the I22D, but not H29A has a profound impact in binding affinities, suggesting that the ABD1 (amino-terminal helix) might also play an important role in actin binding (Fig 5B).

Our mutagenesis study and structural model also suggests that ABD2′ and ABD2 sites of UTRN-ABD could be sufficient for F-actin interaction (Figs 4B and C, and 5B). Therefore, we generated a truncated version called UTRN-mini encompassing amino acids 35–136, which was then tagged with GFP and compared with UTRN-ABD mcherry in U20S cell (Materials and Methods). Co-localization analysis shows that the UTRN-mini versus UTRN-ABD labeling of actin

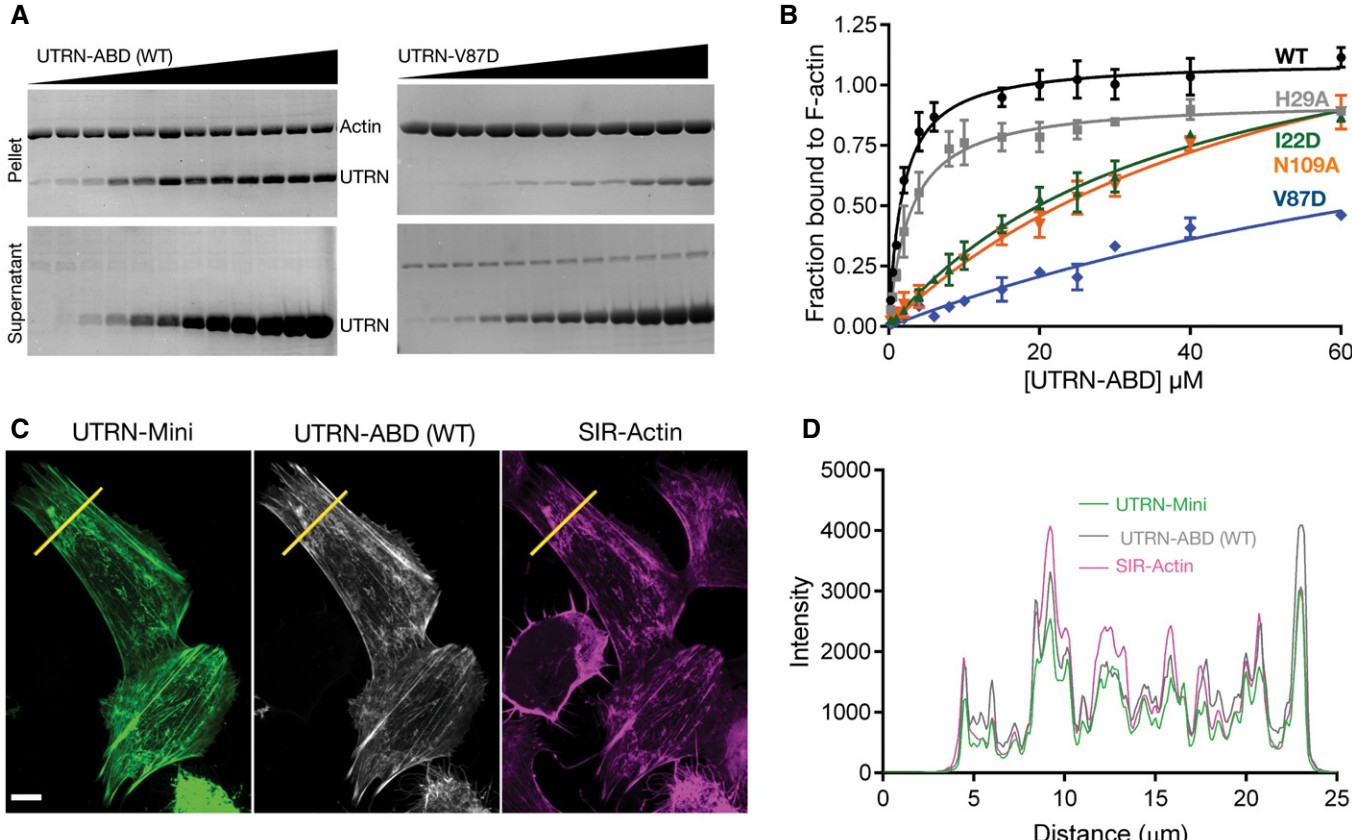

**Figure 5. Mutation analysis of utrophin:F-actin-binding interface.**

A  Representative Coomassie-stained SDS-PAGE gels of UTRN-ABD and UTRN-V87D co-sedimentation with F-actin. Pellet (top) and supernatant (bottom) fractions of individual co-sedimentation reactions of increasing utrophin concentrations, and uncropped gel images of all the co-sedimentation reactions are presented in Appendix Fig S3.

B  Apparent $K_d$ indicated was calculated from the titration data of co-sedimentation assays of utrophin wild-type (1.8 μM) and mutants; H29A (2.8 μM), I22D (38 μM), V87D (> 100 μM), and N109A (55 μM) as indicated. Data points for each concentration were averaged from three independent experiments; error bars represent SD between independent experiments.

C  Confocal images of U2OS cells transfected with GFP tagged UTRN-mini and mCherry-UTRN-ABD, stained with SIR-Actin shows F-actin structures, mainly stress fibers. Scale bar = 5 μm.

D  Co-localization analysis by intensity plot of GFP UTRN-mini, mCherry-UTRN-ABD, and SiR-Actin fluorescence using line scan of the region as indicated by the yellow line in (C).

Source data are available online for this figure.

---

is nearly identical (Fig 5C and D). Together, from our structure and cell labeling studies we conclude that the CH1 domain of utrophin encompassing ABD2 and ABD2′ is sufficient for F-actin interaction.

## Discussion

Using cryoEM, we have determined near atomic resolution structures of phalloidin, lifeAct, and utrophin bound to F-actin structures (Table 1). These structures represent the first high-resolution comparison of most widely used F-actin cellular markers. Similar to most of the known F-actin structural models (Ge *et al*, 2014; Iwamoto *et al*, 2018; Mentes *et al*, 2018; Merino *et al*, 2018; Chou & Pollard, 2019), the markers studied here do not induce any larger deviations in helical parameters of the actin filament (Table 1). By comparing the apo and phalloidin-bound F-actin–ADP structures,

we conclusively show that phalloidin closely resembles the ADP state of actin (Fig 1D). This is in stark contrast to jasplakinolide, which shares the same binding site as phalloidin but causes the D-loop of actin monomers to adopt an open conformation, mimicking the ADP-Pi state (Merino *et al*, 2018). A recent structural study, which determined phalloidin-bound F-actin in different nucleotide states arrived at a similar conclusion (Pospich *et al*, 2020). Phalloidin is widely used in visualizing F-actin in cells; however, phalloidin labeling is currently restricted to only fixed cells. Since phalloidin does not induce any conformational changes, an SiR-actin equivalent of phalloidin fluorescent derivative will be valuable to the actin cytoskeleton community.

Previous studies with utrophin tandem CH1 and CH2 domains, including an X-ray crystal structure (Keep *et al*, 1999), low-resolution electron microscopy models (Moores & Kendrick-Jones, 2000; Galkin *et al*, 2002, 2003), and truncation studies (Singh *et al*, 2017),

portray an ambiguous picture of utrophin:F-actin interaction and the actin-binding sites. In our cryoEM reconstructions, we observe densities corresponding to the CH1 domain (amino acids 18–135) and an additional amino-terminal helix (amino acids 18–30), which was disordered in an earlier crystal structure (Keep *et al*, 1999) (Figs 4A and EV4D). The utrophin CH1 domain architecture and the actin interaction regions are similar to the recently reported mutant FLNaCH1 (filamin) cryoEM structure (Iwamoto *et al*, 2018). However, unlike the FLNaABD, where the CH2 domain showed weaker interaction and poor density map (Iwamoto *et al*, 2018), we could not visualize utrophin CH2 domain in our reconstructions indicating flexibility. Our structural observations of the CH1 domain are consistent with the biophysical characterization of UTRN-ABD, where upon actin binding of the CH1 and CH2 domain gets separated and adopts an open conformation (Lin *et al*, 2011; Broderick *et al*, 2012), which is in contrast to the filamin tandem CH domains. The utrophin-F-actin structure also describes the important actin-binding sites, for example, the ABD2' and ABD1, a helix unique to utrophin CH1 domain. Truncation studies guided by our structural model suggests that amino acids 35–136, i.e., the CH1 domain encompassing ABD2' and ABD2 sites (UTRN-mini), could be sufficient for F-actin interaction (Appendix Fig S4). The UTRN-mini is a 101 amino acid protein that labels F-actin structures in cells (Fig 5C and D), which will occupy lesser footprint on F-actin compared to UTRN-ABD and could be advantageous in actin labeling experiments. The UTRN-mini is in line with biochemical studies of utrophin CH1 domain (Singh *et al*, 2014) and filamin truncation studies, where FLNaCH1 shows similar actin labeling as FLNaABD (Iwamoto *et al*, 2018). Utrophin is also commonly used in biophysical experiments as a load in myosin motility assays (Aksel *et al*,

2015), and the mutations described here will be valuable to biophysicists in fine tuning the load exerted by utrophin in the motor assays. Together, we conclude that the CH1 domain of utrophin is an important element for F-actin interaction and can be used to label F-actin structures in cells (Fig 5C and D).

The lifeAct:F-actin complex cryoEM structure reveals that the lifeAct peptide adopts a 3-turn alpha-helix, as suggested by secondary structure prediction algorithms. The lifeAct interaction with actin is predominantly through hydrophobic contacts, encompassing two neighboring actin monomers. A key feature of this interaction is the overlapping site with D-loop (Figs 2 and 3), from which we hypothesized that lifeAct could sense the closed D-loop conformation, a hallmark of the F-actin–ADP and ADP-Pi state. Our *in vitro* reconstitution experiments using phalloidin and jasplakinolide recapitulates the structural hypothesis that lifeAct detects F-actin in its closed D-loop state. Although our structural and biochemical studies support the importance of the D-loop in F-actin binding, lifeAct can interact with G-actin even more tightly (Riedl *et al*, 2008), which is devoid of the D-loop from the adjacent actin monomer. Thus, a different binding state must exist for G-actin. We predict that in the absence of the D-loop, the charged carboxy-terminus of lifeAct might play a dominant role in mediating additional electrostatic interactions with G-actin. However, in the case of the open D-loop state, the disruption of the hydrophobic pocket could be sufficient to prevent lifeAct's interaction with the F-actin.

Previous cell biology experiments have indicated two limitations of lifeAct: (i) affecting actin polymerization dynamics (Spracklen *et al*, 2014; Courtemanche *et al*, 2016) and (ii) inability to label certain actin structures (Munsie *et al*, 2009; Belin *et al*, 2014). The structural model provided here should help guide the creation of

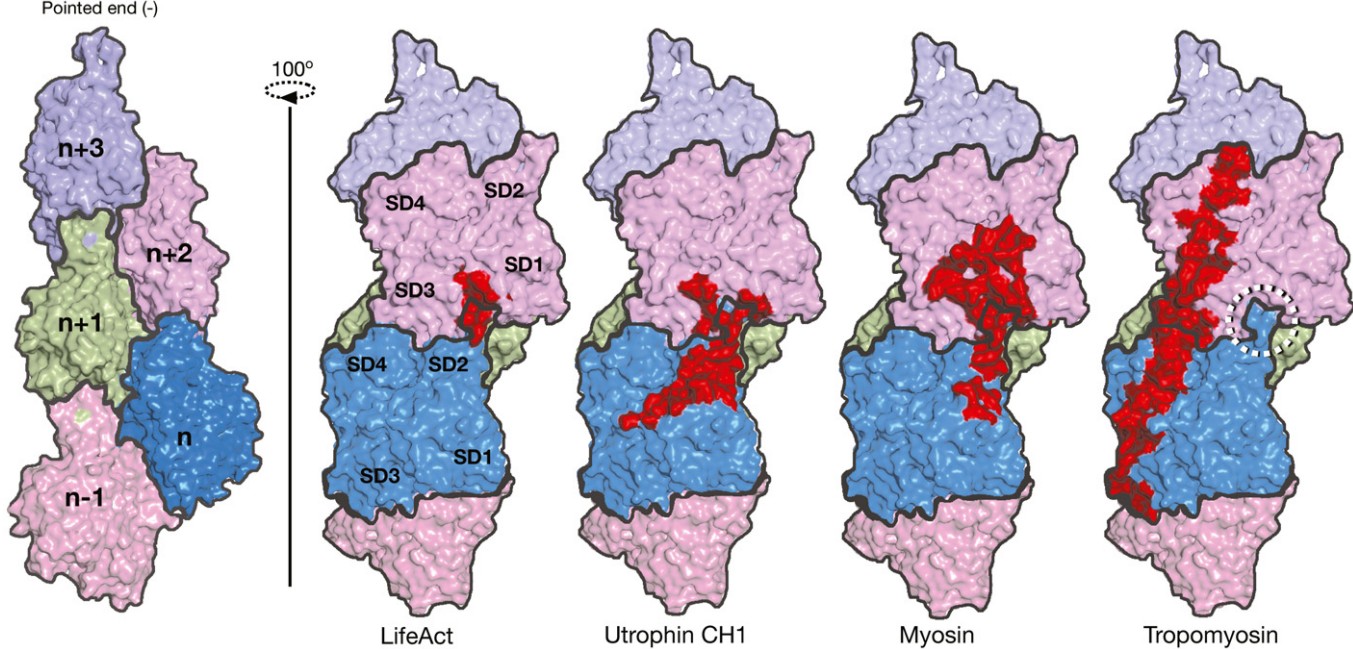

**Figure 6. Interface comparison of UTRN, LifeAct with myosin, and tropomyosin.**

Surface representation of F-actin with five monomers as marked. Footprint (in red) of actin monomers and respective residues interacting with lifeAct, utrophin CH1, myosin, and tropomyosin as indicated and the D-loop is shown circled. The myosin and tropomyosin footprints were derived from PDB IDs 6C1D and 5JLF, respectively.

new lifeAct variants that will mitigate G-actin binding, further improving the suitability of the lifeAct probe. For the second limitation, the most common explanation is that lifeAct might be overlapping with other actin-binding proteins (Fig 6). Based on the experiments described here for lifeAct, we also suggest that the actin structures devoid of lifeAct signal might also reflect different biochemical or alternate D-loop conformations states of actin in cells.

With the recent advances in cryoEM, several actin-binding proteins complexed with F-actin have been characterized (Behrmann *et al*, 2012; Ge *et al*, 2014; Iwamoto *et al*, 2018; Mentes *et al*, 2018). A common theme emerging from these structures is that SD1 and SD2 encompassing the D-loop region of actin is a preferred site for actin-binding proteins (Fig 6). Our lifeAct- and utrophin-bound F-actin structures also show that they overlap with myosin-, cofilin-, and coronin-binding sites (Ge *et al*, 2014; Tanaka *et al*, 2018), but not with that of tropomyosin (Fig 6). Since the D-loop is the sole element that undergoes conformational changes in actin, several reports have proposed that actin-binding proteins might sense the D-loop state. So far coronin and cofilin are known to sense the D-loop in open and closed conformation, respectively (Cai *et al*, 2007; Merino *et al*, 2018). However, upon sensing the closed D-loop conformation, cofilin distorts the F-actin structure resulting in severing of the actin filament (Tanaka *et al*, 2018). Our work here describes the D-loop conformation sensing by lifeAct, which presents lifeAct as the first *bona fide* sensor for the closed D-loop of actin.

In summary, our structural work combined with previous cell biological investigations of various actin markers offers insights into the nature of actin cell markers and their interactions with F-actin, providing an invaluable resource to the actin cytoskeleton community in choosing appropriate actin markers in their investigations.

# Materials and Methods

### DNA constructs and reagents

Human UTRN-ABD (amino acids 1–261) was cloned in pET28a vector with amino-terminal His tag, using GFP-UtrCH (addgene plasmid #26737) as a template. UTRN-ABD mutations were generated in the same vector by Quickchange site-directed mutagenesis (Stratagene). mcherry-UTRN-ABD and eGFP-UTRN-mini (amino acids 35–136) were cloned in mcherry and eGFP pCMV vector, respectively. pLenti-LifeAct-EGFP BlastR was a gift from Ghassan Mouneimne (Addgene plasmid #84383). LifeAct mutations were created in the pLenti-LifeAct-EGFP BlastR using methodology as mentioned for UTRN. Alexafluor-568-phalloidin (Thermo Fisher Scientific Cat. No. A12380) and SiR-actin (Spirochrome Cat. No. Cy-SC001) were purchased. FAM-LifeAct peptides were custom synthesized from LifeTein, USA.

### Protein purification

6xHis-tagged UTRN-ABD and mutants were expressed in *Escherichia coli* Rosetta DE3 strain and induced with 0.25 mM IPTG overnight at 20°C. Bacterial cells were pelleted and resuspended in lysis buffer (50 mM Tris–Cl pH-7.5, 150 mM NaCl, 20 mM Imidazole, 0.1%

Tween-20, and Protease inhibitor cocktail tablet (Roche, Cat. No. 04693159001)). The cells were lysed using sonication, and the lysate was clarified at 39,190 *g* for 30 min. The supernatant fraction containing proteins were loaded on 5 ml His-Trap column (GE Healthcare) and eluted with a linear gradient of elution buffer containing 40–500 mM Imidazole, 500 mM NaCl, and 5 mM beta-mercaptoethanol. The utrophin protein fractions were pooled, concentrated, and loaded on to the Superdex-200 16/600 column, pre-equilibrated with 50 mM Tris–HCl pH-7.5, 150 mM NaCl, 2 mM TCEP, and 0.1% Tween-20. Pure fractions were concentrated using 3 kDa MWCO centrifugal filter unit (Millipore), flash-frozen in liquid nitrogen, and stored at −80°C until use.

### Actin co-sedimentation assays

Actin, purified from chicken breast (*Gallus gallus*) into G buffer (2 mM Tris pH 8, 0.2 mM ATP, 2 mM DTT, 0.2 mM CaCl$_2$) using Spudich laboratory protocol (Pardee & Spudich, 1982). G-actin was polymerized in F-actin buffer (25 mM Tris–Cl, 200 mM KCl, 2 mM MgCl$_2$, and 1 mM ATP) for 2 h at room temperature. Polymerized actin (7.5 µM) was titrated with increasing amounts of UTRN-ABD and mutants in co-sedimentation assay buffer (10 mM Tris–Cl pH 8, 0.5 mM ATP, 0.2 mM DTT, 2 mM MgCl$_2$, and 50 mM KCl). The mixture was incubated at room temperature for 20 min and centrifuged at 100,000 *g* for 30 min in a Beckman TLA-100 rotor. Supernatants were collected, and protein pellets were suspended in equal volume of co-sedimentation assay buffer. The supernatant, pellet, and input samples were loaded in a 10% SDS–PAGE gel for separation. Gels were stained with Coomassie blue and scanned using the iBright FL1000 (Invitrogen). Fiji ImageJ was used for densitometric analysis. For Lifeact-binding assay, polymerized actin (5 µM) was titrated with increasing amount of FAM-lifeAct-17 or FAM-lifeAct-14 in KMEI buffer (50 mM KCl, 1 mM MgCl$_2$, 1 mM EGTA, 10 mM Imidazole pH 7.5). The mixture was incubated at room temperature for 30 min and centrifuged at 350,000 *g* for 30 min in Beckman TLA-100 rotor. Pellet fractions were suspended in equal volume of KMEI buffer, and the bound lifeAct was measured using fluorescence at 485/520 nm using Varioskan Lux multimode microplate (Thermo Fisher Scientific). Data points were fitted to a one-site binding model using Prism software (GraphPad) to calculate the apparent binding affinity and stoichiometry as described earlier (Riedl *et al*, 2008; Singh *et al*, 2014).

### Sample and grid preparation for cryoEM

Freshly prepared G-actin was used for polymerization. For F-actin-phalloidin complex, actin was polymerized in F-actin buffer (10 mM Imidazole pH 7.4, 200 mM KCl, 2 mM MgCl$_2$, 1 mM ATP) at room temperature for 2 h and then Alexafluor-568-phalloidin (1/2 ratio) was mixed and incubated overnight at 4°C. For F-actin-utrophin, F-actin-ADP-Apo and F-actin-lifeAct complex, polymerization was induced with KMEI buffer (50 mM KCl, 1 mM MgCl$_2$, 0.5 mM ATP, 1 mM EGTA, and 10 mM Imidazole pH 7.5) at 4°C overnight.

In the case of F-actin–phalloidin and F-actin-ADP-Apo, we applied 3.0–3.5 µl of sample onto a freshly glow-discharged Quantifoil Au 1.2/1.3, 300 mesh grids. For F-actin–utrophin, 5–8 µM of F-actin was applied on to Au 1.2/1.3 grid and then 4–5 molar excess of utrophin was mixed to it and incubated for 30–60 s at > 95%

humidity, then blotted for 3–3.5 s. For F-actin–lifeAct, Au 0.6/1.0 grid was used and same amount of F-actin as above with excess molar concentration of lifeAct peptide was used for sample preparation. Grids were prepared with Thermo Fisher Scientific Vitrobot Mark IV. All grids were incubated for 30–60 s at > 95% humidity, then blotted for 3–3.5 s. Immediately after blotting, the grids were plunge-frozen in liquid ethane.

### CryoEM data collection

The datasets were collected on FEI Titan Krios G3 transmission electron microscope equipped with a FEG at 300 kV with the automated data collection software EPU (Thermo Fisher Scientific) at the National CryoEM facility, Bangalore. Images of the F-actin–phalloidin, F-actin-UTRN-ABD and F-actin-LifeAct were collected with a Falcon III detector operating in linear mode at a nominal magnification of 59,000× and a calibrated pixel size of 1.38 Å, while the F-actin–ADP (apo) was collected at a nominal magnification of 75,000× and a pixel size of 1.08 Å. In all cases, we acquired one image per grid hole. Table 1 contains the details on exposure time, frame number, and electron dose for all the datasets.

### Data processing and model building

Unaligned frame images were manually inspected and evaluated for ice and filament quality. After manual removal of bad images, the remaining movie micrographs were motion corrected with either by Unblur (Grant & Grigorieff, 2015) or by algorithm inbuilt in Relion 3.0 (Scheres, 2012; He & Scheres, 2017). CTF estimation was performed with GCTF (Zhang, 2016) on the full-dose weighted motion-corrected sums. For all the datasets, filaments were manually selected and processed with Relion 3.0 (He & Scheres, 2017). We used a box size of 256 pixels for phalloidin-, utrophin- and lifeAct-bound F-actin dataset and 320 pixels for the F-actin–ADP (apo) with the interbox distance of 27.5 Å for extraction of segments. Subsequently, 2D classification in Relion 3.0 was used to remove bad segments. To further remove the partially decorated filament, we did helical 3D classification using F-actin (EMDB-1990) as a reference, which was low-pass filtered to 30–35 Å, to avoid reference bias. The best decorated 3D classes were combined and used for refinement using same reference as above starting with the sampling rate of 1.8°. All refinement steps were performed with soft mask containing 75–80% of the filament. To further improve the maps, we performed CTF refinement and Bayesian polishing of lifeAct- and utrophin-bound F-actin datasets (Zivanov et al, 2018). The polished particles were subsequently refined, and the resolutions for all datasets were estimated with mask and post-processing option in Relion 3.0. Local resolution of the maps were estimated with Resmap (Kucukelbir et al, 2014).

We used F-actin structure (PDB-6BNO) as a starting atomic model in Chimera (Pettersen et al, 2004) to fit the F-actin model in the map for all datasets. Phalloidin coordinates were used from PDB-6D8C (Iwamoto et al, 2018). For F-actin-utrophin complex, utrophin CH1 domain from crystal structure PBD-1QAG was used as the starting model (Keep et al, 1999). Coot (Emsley et al, 2010) was used for model building for all datasets and real space refined using Phenix (Adams et al, 2010; Afonine et al, 2018). All structural models were validated in MolProbity (Williams et al, 2018) and

PDBe site. Figures were generated using Pymol (DeLano, 2002), Chimera (Goddard et al, 2007), and Coot (Emsley et al, 2010) software programs.

### Cell imaging

Wild-type U2OS cells were obtained as a gift from Prof. Satyajit Mayor's laboratory, NCBS, Bangalore, India. For all the experiments, U2OS cells were cultured in McCoy's 5A (Sigma Aldrich, M4892) media supplemented with 2.2 g/l sodium bicarbonate, 10% fetal bovine serum (FBS) and 1× PenStrep (cat. no. 15-140-122 Gibco Fisher Scientific) in a humidified 37°C incubator with 5% carbon dioxide. Around 20,000 to 30,000 cells were seeded in ibidi glass-bottom dishes (Cat. No. 81218, Ibidi), and transfection was carried out at 60–70% confluency of the cells with total of 1 μg of plasmid DNA used for transfection. All the transfection experiments were carried out with jet prime transfection reagent (cat. no. 114-15 polypus transfection) as described in manufacturer's protocol. Five hundred nanogram of UTRN-mini construct (UTRN 35-136) tagged with N-terminus EGFP were co-transfected with 500 ng plasmid of mcherry tagged UTRN-ABD in the 10% serum containing media. The transfection media was changed with fresh media after 4–6 h of transfection. Cells were imaged after 24 h of transfection with SiR-actin (Cat. no. CY-SC001 Spirochrome kit; Cytoskeleton, Inc) staining in the complete media. All the images were obtained at 60× oil objective (1.42NA) with 2,048*2,048 frame size and 0.5–1 μm optical sections on FV3000 Olympus confocal microscope equipped with 488, 561, and 640 laser for GFP, cy3, and cy5 channels, respectively. All the images obtained through Olympus software and were analyzed on Fiji ImageJ.

### *In vitro* actin labeling assay and TIRF microscopy

Flow chambers of ~ 10 μl volume were prepared using double-sticky tape, coverslips, and cover glass. The flow chamber was incubated with Protein G (Sigma, Cat. No. 08062) for 10 min followed by anti-his antibody (Sigma, Cat. No. 11922416001) for another 10 min. After washing with KMEI buffer without ATP, UTRN-ABD-6xHis was flowed to attach actin filament with the coverslip. The Phalloidin-Actin-568 and SiR-actin 640 F-actin were prepared separately and added together with different concentration of FAM-lifeAct. The mixture was incubated in tube for 5–10 min and flowed in the chamber for visualization. Flow chambers were imaged at 100× oil objective 1.49NA under the total internal reflection mode using Nikon Ti2 H-TIRF system with 488, 561, and 640 laser lines. The images were acquired for all the three channels near glass surface sequentially with appropriate spectrum filter sets using s-CMOS camera (Hamamatsu Orca Flash 4.0) controlled by NIS-elements software. All images and data were analyzed using Fiji ImageJ software.

## Data availability

CryoEM maps and coordinates are deposited in EMDB and PDB under following code; PDB 7BT7 (https://www.ebi.ac.uk/pdbe/entry/pdb/7bt7) and EMD-30171 (http://www.ebi.ac.uk/pdbe/entry/EMD-30171) for F-actin:ADP apo, PDB 7BTI (https://www.ebi.ac.uk/pdbe/entry/pdb/7bti) and EMD-30179 (http:/ www.ebi.ac.uk/

pdbe/entry/EMD-30179) for F-actin:ADP-phalloidin, PDB 7BTE (https://www.ebi.ac.uk/pdbe/entry/pdb/7bte) and EMD-30177 (http://www.ebi.ac.uk/pdbe/entry/EMD-30177) for F-actin:lifeAct, and PDB 6M5G (https://www.ebi.ac.uk/pdbe/entry/pdb/6m5g) and EMD-30085 (http://www.ebi.ac.uk/pdbe/entry/EMD-30085) for F-actin:utrophin. The datasets for this study are available from the corresponding author upon reasonable request.

**Expanded View** for this article is available online.

## Acknowledgements

The authors wish to thank Dr. Jim Spudich and members of the Sirajuddin laboratory for comments on manuscript. The authors acknowledge the National CryoEM Facility at Bangalore Life Science Cluster and funding by B-life grant from Department of Biotechnology (DBT/PR12422/MED/31/287/2014) and the Central Imaging and Flow Facility (CIFF) at NCBS/inStem/CCAMP campus. Financial support from the DBT-RA Program (Fellowship No. 40796915460) in Biotechnology and Life Sciences is gratefully acknowledged by A.K. This work was funded by NCBS-TIFR/DAE core grants and SERB-Ramanujan Fellowship to KRV. This work was supported by the inStem core grants from Department of Biotechnology, India, DBT/Wellcome Trust—India Alliance Fellowship [grant number IA/I/14/2/501533], CEFIPRA (5703-1) and EMBO Young Investigator award to MS.

## Author contributions

AK and MS conceived the project. AK, MGJ and KRV performed cryoEM work and biochemical analysis. AK, SK and MS performed cell biology and TIRF experiments. KRV and MS supervised the project. AK and MS wrote the paper and all authors commented on the manuscript.

## Conflict of interest

The authors declare that they have no conflict of interest.

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
