## [Review Process File · The EMBO Journal]

Structural insights into actin filament recognition by commonly used cellular actin markers

Archana Kumari, Shubham Kesarwani, Manjunath Javoor, Kutti Vinothkumar, and Minhajuddin Sirajuddin

DOI: [10.15252/embj.2019104006](https://doi.org/10.15252/embj.2019104006)

Corresponding author(s): Minhajuddin Sirajuddin (minhaj@instem.res.in), Kutti Vinothkumar (vkumar@ncbs.res.in)

Review Timeline:

Submission Date:	13th Nov 19
Editorial Decision:	17th Jan 20
Revision Received:	2nd Apr 20
Editorial Decision:	5th May 20
Revision Received:	11th May 20
Accepted:	18th May 20

Editor: Ieva Gailite

Transaction Report:

Thank you for submitting your manuscript for consideration by the EMBO Journal. I apologise for the protracted review process due to delays in review submission over the holiday period. We have now received two referee reports on your manuscript, which are included below for your information.

As you will see from the comments, while reviewer #2 appreciates the study and supports its publication here after a revision, reviewer #1 is more critical regarding the broader novelty of the study in the context of previously published structures, the depth of the provided characterisation and the broader interest of the study to the scientific community. Due to these disparate assessments, I have asked advice from an external scientific advisor with expertise in actin cytoskeleton dynamics, who is supportive of publication of a revised version if the study could be substantially extended along the lines requested by both reviewers, including the points 3 and 4 from reviewer #1 and point 1 from reviewer #2.

Based on these assessments, I would like to invite you to submit a revised version of your manuscript in response to reviewers' comments. I realise that the point 3 from reviewer #1 on design of optimized LifeAct molecules is rather far reaching, therefore please contact me if you would like to discuss feasibility of this or any other aspects of revision. I should add that it is The EMBO Journal policy to allow only a single major round of revision and that it is therefore important to resolve the main concerns at this stage.

We generally allow three months as standard revision time, but an extension up to six months is possible for more extensive revisions. Please contact us in advance if you would need an additional extension. As a matter of policy, competing manuscripts published during this period will not negatively impact on our assessment of the conceptual advance presented by your study. However, please contact me as soon as possible upon publication of any related work in order to discuss how to proceed.

Referee #1:

Kumari et al. describe cryoEM structures of widely used actin probes: phalloidin, Lifeact and the utrophin actin-binding domain (UTRN-ABD). The rationale of the study was to provide structural information that would help researchers in the field to select a suitable actin probe for their cell biological studies.

The cryoEM structures and accompanied biochemical work appear of good technical quality. However, although the structures of these actin probes bound to F-actin are interesting, the findings presented in the manuscript are partially redundant with earlier publications. Moreover, the study does not provide such novel actin probes or information that would significantly benefit researches when visualizing actin structures in cells. Thus, at least in its present form, this study may be better suited for publication in a more specialized journal.

Major points:

1. The structure of phalloidin bound to F-actin was previously reported in other publications (e.g.

Mentes et al 2018; Iwamoto et al 2018). However, the authors did not present a careful comparison of their F-actin/phalloidin structure vs. the published structures, and thus it remains unclear what new we can learn from the structure presented here. Moreover, important biological questions, such as how phalloidin stabilizes F-actin or how it might affect binding of actin-binding proteins were not analysed or discussed.

2. The structure of UTRN-ABD to F-actin appears to be very similar to the recently published FLNa-ABD/F-actin structure, and again without careful comparison of these structures, it is somewhat unclear what new we can learn from the UTRN-ABD structure presented here. Moreover, previous biochemical studies on UTRN-ABD, as well as filamin truncation studies, already demonstrated that the first CH domain harbours the strong actin-binding site.

3. The obtained Lifeact/F-actin structure is the most novel finding of the manuscript. The authors also performed mutagenesis studies and in vitro TIRF experiments to more carefully analyse the preference of Lifeact for different states of actin filaments. However, the authors did not apply this information for designing new actin probes. The impact of this study would be greatly increased, if the authors could design new Lifeact variants, which would e.g. be more specific towards actin filaments.

4. By comparing the binding of Lifeact to phalloidin- and JASP-stabilized actin filaments, the authors conclude that Lifeact prefers the ADP-state of actin. However, this conclusion is based on indirect observations. Thus, the authors should study this more directly by determining the affinities of Lifeact for ADP-, ADP-Pi-, and ATP-F-actin by using e.g. AMPPNP and ADP with inorganic phosphate (see e.g. Chou and Pollard, 2019).

Minor points:

1. There are some overstatements in the text. For example, lines 17-20: the 3.6 - 4.2 Å structures presented here are not the first 'high resolution' structures of widely used actin probes, because structures of phalloidin and JASP in complex with F-actin were already reported.

2. Lines 118-119: The structure presented here is an average model of phalloidin/F-actin. Thus, I am not certain that the authors can conclude (from the structure of this resolution) that the stabilising effect of phalloidin of actin filaments is not due to contacts to all three actins surrounding phalloidin.

3. Line 123-124: Is the resolution sufficient to reliably build the indicated side chains and compare the effects of phalloidin? The authors should show electron densities of the side chains in supplementary information.

4. Lines 296-300: This is an interesting statement, which requires further experimental analysis. The authors could utilise mutants/structural information to test if any of the mutants displays better preference towards filamentous or monomeric actin. From the figures, it is also not clear what is the binding interface of Lifeact on G-actin, and this should be studied in more detail. Better analysis of the structure, combined with additional mutagenesis and biochemical work may also allow the authors to design F-actin/G-actin specific Lifeact variants.

5. Fig 1A: Polarity of the filament appears incorrect.

6. Fig 1D: PDB-code for JASP/F-actin should be included in the figure legend.

7. Figs. 2B and 2C: This presentation is quite messy and not particularly informative. Something similar to 1C would be better. Authors should also show the electron density in a similar manner as in Fig. 1B.
8. Fig 2D. Scale bars should be included in the figure, and specified in the legend. The text in line plots axes is not readable.
9. Fig 4A: Polarity of the filament is incorrect.
10. Fig 4E: Why not show the electron densities here?
11. Legend to Fig 5C: Measure for the scale bar is missing.
12. Fig 6: Polarity of the filament is incorrect.

Referee #2:

The manuscript by Archana Kumari and co-workers describes the interaction of cellular actin markers with the actin filament. Three commonly used actin markers, phalloidin, lifeAct and utrophin, were used and the complex structures bound to F-actin determined by cryo electron microscopy at molecular resolutions of 3.6 to 4.2 Angstrom, respectively. In addition, an F-actin apo-structure was determined at 3.8 Ang resolution. All actin filaments contain ADP. The molecular basis of F-actin marker binding is compared for all three ligands and the binding interfaces are set in comparison to cellular actin binding factors as myosin and tropomyosin, showing that the binding site for LifeAct and Utrophin on F-actin is mutually exclusive and similar to the Myosin binding site but different from the tropomyosin binding interface. It is interesting to note that the three actin markers are of very different nature as the small natural compound phalloidin (Mw 789 Da), the helical peptide lifeAct of 17 aa, and the globular Utrophin protein, whose actin binding site encompasses 2 Calponin-homology (CH) domains of together 29 kDa.

The authors find that Phalloidin interacts with two but not three actin molecules in the filament interface of the two strains, but is at least 5 Ang apart from the n+2 monomer, an observation which is similar for Jasplakinolide. Binding of LifeAct to F-actin covers two actin molecules along one filament strain (n and n-2) and requires a conformation of the D-loop which is associated with a closed conformation found in the ADP-bound state of actin. Mutational analyses were performed to probe the effect of single residues in lifeAct on the staining of F-actin in cells confirming the structural data. Similarly, the CH1 domain of Utrophin was engineered to design a minimal actin binding domain, as the structure revealed binding to actin only for the first CH domain while the C-terminal CH2 domain was not resolved in the final EM reconstructed map. Here again, important binding residues in Utrophin derived from the complex structure were confirmed by co-sedimentation assays of single point mutations and actin filament staining microscopy in cells.

The manuscript is a comprehensive resource of F-actin binding markers that systematically analyses the features of these three commonly used tools. Therefore, the topic is of interest to many researchers and may gain high visibility. However, to my opinion the manuscript needs severe text editing, particularly the comma placement appears strange in many cases (e.g. first sentence of the introduction). I have a couple of suggestions to strengthen the study.

In support of the Actin-ADP bound state proposed for LifeAct binding, could the authors use a recombinant filament of actin with a non-hydrolysable ATP analog, e.g. AppNHp, to test LifeAct binding? E.g. from a quantification of co-sedimentation assays using lifeAct-GFP, similarly as performed for Utrophin in Fig. 5A,B?

To this reviewers' opinion, the colours of the actin filament display in Figs 1, 2, 4, 6 is chosen in a suboptimal manner. The blue and green is too bright, and particularly the differentiation in the blue and green tones is too small. Choosing colours with a higher degree of grey/white tones (as pastel) would give more contrast and depth in the surface display.

Table 1: I suggest renaming the second column "F-actin-ADP" into "apo F-actin" or anything similar. All four structures contain ADP as supposed from line 691: Ligands. So "Apo" is a much better differentiation for this structure compared to the complexed filaments. In addition, I suggest to provide the PDB and EMD codes in the last line of this table for better correlation to the data.

Results, line 109: please name the resolution to which the two structures were resolved, same as given for lifeAct (line 141) and Utrophin (line 191).

The subdomain composition of actin (line 146) is not introduced, although they are marked in several figures (1+6).

Summary of Reviewer response:

We thank both the reviewers for taking valuable time in providing constructive feedback of our manuscript. A converging comment is to measure the LifeAct affinity towards different nucleotide states of F-actin. Our experiments show that LifeAct has no difference between ADP, ADP-Pi and AMPPNP F-actin. This because in all these nucleotide states the D-loop remains in a closed conformation (Chou and Pollard 2019 & Merino et al 2019), which is also the preferred conformation for LifeAct. A recent preprint describing LifeAct:F-actin structure (Beyly et al 2020 bioRxiv) has shown ADP-Pi bound F-actin with LifeAct, which is in line with our biochemical measurements. Using jasplakinolide, we have forced the D-loop to remain in open conformation and this experiment shows LifeAct's preference towards the closed conformation. This result is an indirect observation, which the reviewer 1 has rightly pointed. Therefore, we have modified our sentence to “lifeAct specifically recognizes closed D-loop conformation i.e., ADP-Pi or ADP states of F-actin” in the abstract and as well as in the conclusion section. The other major comment from reviewer 1 is whether we can generate new lifeAct variants. From our structure we have designed 14 amino acid LifeAct peptide, which binds well to F-actin in cells and in our biochemical assay. However, a much more comprehensive study will be required to probe new lifeAct variants, which is well beyond the scope of our manuscript. We have also fully addressed other minor comments and again thank the reviewers for suggestions in improving our manuscript.

Referee #1:

Kumari et al. describe cryoEM structures of widely used actin probes: phalloidin, Lifeact and the uthropin actin-binding domain (UTRN-ABD). The rationale of the study was to provide structural information that would help researchers in the field to select a suitable actin probe for their cell biological studies.

The cryoEM structures and accompanied biochemical work appear of good technical quality. However, although the structures of these actin probes bound to F-actin are interesting, the findings presented in the manuscript are partially redundant with earlier publications. Moreover, the study does not provide such novel actin probes or information that would significantly benefit researches when visualizing actin structures in cells. Thus, at least in its present form, this study may be better suited for publication in a more specialized journal.

Major points:

1. The structure of phalloidin bound to F-actin was previously reported in other publications (e.g. Montes et al 2018; Iwamoto et al 2018). However, the authors did not present a careful comparison of their F-actin/phalloidin structure vs. the published structures, and thus it remains unclear what new we can learn from the structure presented here. Moreover, important biological

questions, such as how phalloidin stabilizes F-actin or how it might affect binding of actin-binding proteins were not analysed or discussed.

The currently available phalloidin structures (e.g., Iwamoto et al 2018 and Menten et al 2018) have either filamin or myosin bound to F-actin, which overlaps with the D-loop. Therefore, a clear conclusion about the D-loop state could not be inferred. Our phalloidin bound actin structure is perhaps the first without any additional proteins thus allowing us to conclusively report the D-loop confirmation. We have discussed this point in the Results (line 107) and Discussion.

We have also added additional points related to application of phalloidin in labeling F-actin (line 295).

2. The structure of UTRN-ABD to F-actin appears to be very similar to the recently published FLNa-ABD/F-actin structure, and again without careful comparison of these structures, it is somewhat unclear what new we can learn from the UTRN-ABD structure presented here. Moreover, previous biochemical studies on UTRN-ABD, as well as filamin truncation studies, already demonstrated that the first CH domain harbours the strong actin-binding site.

Our UTRN:F-actin structure has conclusively showed that CH1 domain is sufficient, previous works have hinted at this point but failed to generate a robust UTRN-CH1 domain. Based on the structure, we show that a minimal utrophin called UTRN-mini (35-136 amino acids) is sufficient for actin binding and can be used for labelling F-actin in cells (Figure 5C).

In the current revision we have also added the comparison of Filamin ABD, UTRN x-ray and EM structures (Appendix Figure S4, Supplement information).

3. The obtained Lifeact/F-actin structure is the most novel finding of the manuscript. The authors also performed mutagenesis studies and in vitro TIRF experiments to more carefully analyse the preference of Lifeact for different states of actin filaments. However, the authors did not apply this information for designing new actin probes. The impact of this study would be greatly increased, if the authors could design new Lifeact variants, which would e.g. be more specific towards actin filaments.

From the structure of LifeAct we have truncated the LifeAct to 14 aa and found that it labels F-actin in cells and has similar affinity (as lifeAct-17) towards F-actin (Figure 2D).

4. By comparing the binding of Lifeact to phalloidin- and JASP-stabilized actin filaments, the authors conclude that Lifeact prefers the ADP-state of actin. However, this conclusion is based on indirect observations. Thus, the authors should study this more directly by determining the affinities of Lifeact for ADP-, ADP-Pi-, and ATP-F-actin by using e.g. AMPPNP and ADP with inorganic phosphate (see e.g. Chou and Pollard, 2019).

We have measured the affinity of LifeAct towards F-actin ADP, ADP-Pi and AMPPNP and found practically no difference in affinities between these different states. Studies by Chou and Pollard, 2019 and Merino et al 2019, have conclusively showed that D-loop exists in a closed conformation

in ADP-Pi, AMPPNP and ADP states. Merino et al 2019 showed that the D-loop open conformation can be achieved either by having transition state analog BeFx or jasplakinolide, which we have used in our binding assays and showed LifeAct preference towards closed D-loop conformation.

A recent preprint reporting F-actin:LifeAct structure (Beyly et al 2020 bioRxiv) has also provided structural evidence that LifeAct can bind to ADP-Pi F-actin. This supports our argument about F-actin nucleotide states versus LifeAct affinity measurement.

We agree with the reviewer comment that this is an indirect observation, therefore we have modified our conclusion accordingly in the abstract (line 21) and in discussion (line 307).

Minor points:

1. There are some overstatements in the text. For example, lines 17-20: the 3.6 - 4.2 Å structures presented here are not the first 'high resolution' structures of widely used actin probes, because structures of phalloidin and JASP in complex with F-actin were already reported.

We thank the reviewer in pointing out this, we have changed the sentences to “...providing a comprehensive high-resolution structural comparison of widely used actin markers...” (line 19)

2. Lines 118-119: The structure presented here is an average model of phalloidin/F-actin. Thus, I am not certain that the authors can conclude (from the structure of this resolution) that the stabilising effect of phalloidin of actin filaments is not due to contacts to all three actins surrounding phalloidin.

In order to support our claim of contacts we have added Appendix Figure S1E (supplement information), showing the residues with maps and distance between them and phalloidin.

3. Line 123-124: Is the resolution sufficient to reliably build the indicated side chains and compare the effects of phalloidin? The authors should show electron densities of the side chains in supplementary information.

We included the residues and maps in Appendix Figure S1, supplement information.

4. Lines 296-300: This is an interesting statement, which requires further experimental analysis. The authors could utilise mutants/structural information to test if any of the mutants displays better preference towards filamentous or monomeric actin. From the figures, it is also not clear what is the binding interface of Lifeact on G-actin, and this should be studied in more detail. Better analysis of the structure, combined with additional mutagenesis and biochemical work may also allow the authors to design F-actin/G-actin specific Lifeact variants.

We have included a truncated version of LifeAct (14LA), which has similar affinity towards F-actin compared to full LifeAct 17 amino acids (17LA). The suggested experiments to design F-actin/G-actin specific LifeAct variants is currently beyond the scope of this manuscript, especially under

the current lockdown situation. We hope that our structural work will inspire others researchers in designing LifeAct variants specific towards actin filaments.

5. Fig 1A: Polarity of the filament appears incorrect.
6. Fig 1D: PDB-code for JASP/F-actin should be included in the figure legend.
7. Figs. 2B and 2C: This presentation is quite messy and not particularly informative. Something similar to 1C would be better. Authors should also show the electron density in a similar manner as in Fig. 1B.
8. Fig 2D. Scale bars should be included in the figure, and specified in the legend. The text in line plots axes is not readable.
9. Fig 4A: Polarity of the filament is incorrect.
10. Fig 4E: Why not show the electron densities here?
11. Legend to Fig 5C: Measure for the scale bar is missing.
12. Fig 6: Polarity of the filament is incorrect.

We thank the reviewer for pointing out these errors, we have corrected all these in our revised manuscript according the minor suggestions from point 5 – 12.

Referee #2:

The manuscript by Archana Kumari and co-workers describes the interaction of cellular actin markers with the actin filament. Three commonly used actin markers, phalloidin, lifeAct and utrophin, were used and the complex structures bound to F-actin determined by cryo electron microscopy at molecular resolutions of 3.6 to 4.2 Angstrom, respectively. In addition, an F-actin apo-structure was determined at 3.8 Ang resolution. All actin filaments contain ADP. The molecular basis of F-actin marker binding is compared for all three ligands and the binding interfaces are set in comparison to cellular actin binding factors as myosin and tropomyosin, showing that the binding site for LifeAct and Utrophin on F-actin is mutually exclusive and similar to the Myosin binding site but different from the tropomyosin binding interface. It is interesting to note that the three actin markers are of very different nature as the small natural compound phalloidin (Mw 789 Da), the helical peptide lifeAct of 17 aa, and the globular Utrophin protein, whose actin binding site encompasses 2 Calponin-homology (CH) domains of together 29 kDa.

The authors find that Phalloidin interacts with two but not three actin molecules in the filament interface of the two strains, but is at least 5 Ang apart from the n+2 monomer, an observation which is similar for Jasplakinolide. Binding of LifeAct to F-actin covers two actin molecules along one filament strain (n and n-2) and requires a conformation of the D-loop which is associated with a closed conformation found in the ADP-bound state of actin. Mutational analyses were performed to probe the effect of single residues in lifeAct on the staining of F-actin in cells confirming the

structural data. Similarly, the CH1 domain of Utrophin was engineered to design a minimal actin binding domain, as the structure revealed binding to actin only for the first CH domain while the C-terminal CH2 domain was not resolved in the final EM reconstructed map. Here again, important binding residues in Utrophin derived from the complex structure were confirmed by co-sedimentation assays of single point mutations and actin filament staining microscopy in cells.

The manuscript is a comprehensive resource of F-actin binding markers that systematically analyses the features of these three commonly used tools. Therefore, the topic is of interest to many researchers and may gain high visibility. However, to my opinion the manuscript needs severe text editing, particularly the comma placement appears strange in many cases (e.g. first sentence of the introduction). I have a couple of suggestions to strengthen the study.

In support of the Actin-ADP bound state proposed for LifeAct binding, could the authors use a recombinant filament of actin with a non-hydrolysable ATP analog, e.g. AppNHp, to test LifeAct binding? E.g. from a quantification of co-sedimentation assays using lifeAct-GFP, similarly as performed for Utrophin in Fig. 5A,B?

Our LifeAct titration experiments with AMPPNP and ADP-Pi F-actin did not show the differences that were observed in the TIRF experiments. This is because of the structural similarities in D-loop conformation between these nucleotide states, which we have explained in the summary above.

To this reviewers' opinion, the colours of the actin filament display in Figs 1, 2, 4, 6 is chosen in a suboptimal manner. The blue and green is too bright, and particularly the differentiation in the blue and green tones is too small. Choosing colours with a higher degree of grey/white tones (as pastel) would give more contrast and depth in the surface display.

We have changed to a milder yet contrasting color combination as suggested by this reviewer.

Table 1: I suggest renaming the second column "F-actin-ADP" into "apo F-actin" or anything similar. All four structures contain ADP as supposed from line 691: Ligands. So "Apo" is a much better differentiation for this structure compared to the complexed filaments. In addition, I suggest to provide the PDB and EMD codes in the last line of this table for better correlation to the data.

We thank the reviewer for the suggestion we have changed the heading to 'apo F-actin-ADP'

Results, line 109: please name the resolution to which the two structures were resolved, same as given for lifeAct (line 141) and Utrophin (line 191).

The resolution for respective structures has been added (line 142 for lifeAct) and (line 206 for utrophin)

The subdomain composition of actin (line 146) is not introduced, although they are marked in several figures (1+6).

We thank the reviewer for pointing this, we have added a sentence related to subdomains of actin in introduction (line 31).

Thank you for submitting a revised version of your manuscript. Your study has now been seen by both original referees, who find that their main concerns have been addressed and are now broadly in favour of publication of the manuscript. There now remain only a few mainly editorial issues that have to be addressed before I can extend formal acceptance of the manuscript:

1. According to the request from reviewer #2, please add the data on LifeAct affinity towards nucleotide-bound F-actin to the Appendix.
2. Our publisher has done their pre-publication check on your manuscript. When you log into the manuscript submission system you will see the file "Data Edited Manuscript". Please take a look at the word file and the comments regarding the figure legends and respond to the issues. Please also use this version when you resubmit the revised version with the marked changes
3. Please submit up to five keywords.
4. Please restructure your manuscript according to our format: Materials and Methods should follow Discussion. Please move Data Availability section at the end of Materials and Methods. Please place Acknowledgements, Author Contributions and Conflict of Interest sections after Materials and Methods.
5. Please add resolvable links to the publicly available datasets in the Data Availability section. The Data Availability section should follow the format indicated in our Author Guidelines:
<https://www.embopress.org/page/journal/14602075/authorguide#dataavailability>
6. Please rename Table EV1 into Table 1 in the legend and in the callouts in the manuscript text.
7. Thank you for submitting source data for your manuscript. Please arrange the data in one file per figure. Please check if the data correspond to the correct figure panels - Figure 3C data do not appear to fit to the type of data shown in the figure and rather to correspond to Figure 2D.
8. Papers published in The EMBO Journal are accompanied online by a 'Synopsis' to enhance discoverability of the manuscript. It consists of A) a short (1-2 sentences) summary of the findings and their significance, B) 2-3 bullet points highlighting key results and C) a synopsis image that is 550x300-600 pixels large (width x height, jpeg or png format). You can either show a model or key data in the synopsis image. Please note that the size is rather small and that text needs to be readable at the final size. Please send us this information along with the revised manuscript.

Please let me know if you have any further questions regarding any of these points. You can use the link below to upload the revised files.

Thank you again for giving us the chance to consider your manuscript for The EMBO Journal. I am looking forward to receiving the final version.

Referee #1:

The authors have now satisfactorily addressed my previous concerns.

Referee #2:

In the revised version of the manuscript the authors improved the description of the three markers for the actin filament significantly. The text reads much better and the presentation of phalloidin, LifeAct, and utrophin binding to the actin filament is clear. It is a pity that the authors did not show the primary data of their affinity measurements of LifeAct towards ADP, ADP-Pi or AppNHp loaded F-actin, at least I could not find them. The authors state in the reply letter that they found practically no difference in affinities between these different states, in line with very recent descriptions on bioRxiv. The changed colors for the actin molecules in the filament display are not really an improvement to this reviewer's opinion. I rather thought of grey tones, but anyways, this is a very minor issue. The findings of LifeAct binding to F-actin are very timely, which is why I suggest publication of this study.

The authors performed the requested changes.

Thank you for addressing the final minor issues in the revised manuscript. I am now pleased to inform you that your manuscript has been accepted for publication.

Corresponding Author Name: Minhajuddin Sirajuddin

Journal Submitted to: The EMBO Journal

Manuscript Number: EMBOJ-2019-104006